# Intrinsic and Microenvironmental Drivers of Glioblastoma Invasion

**DOI:** 10.3390/ijms25052563

**Published:** 2024-02-22

**Authors:** Emerson De Fazio, Matilde Pittarello, Alessandro Gans, Bikona Ghosh, Hasan Slika, Paolo Alimonti, Betty Tyler

**Affiliations:** 1Department of Medicine, Vita-Salute San Raffaele University School of Medicine, 20132 Milan, Italy; e.defazio@studenti.unisr.it (E.D.F.); p.alimonti@studenti.unisr.it (P.A.); 2Department of Medicine, Humanitas University School of Medicine, 20089 Rozzano, Italy; matilde.pittarello@gmail.com; 3Department of Neurology, University of Milan, 20122 Milan, Italy; alessandro.gans@unimi.it; 4School of Medicine and Surgery, Dhaka Medical College, Dhaka 1000, Bangladesh; bikonaghosh.01@gmail.com; 5Hunterian Neurosurgical Laboratory, Department of Neurosurgery, Johns Hopkins University School of Medicine, Baltimore, MD 21231, USA; hslika1@jh.edu; 6Department of Neurosurgery, Brigham and Women’s Hospital, Harvard Medical School, Boston, MA 02115, USA

**Keywords:** glioma, invasion, microenvironment, molecular mechanisms, neuronal niche, astrocytes, immune cells, therapeutic targeting

## Abstract

Gliomas are diffusely infiltrating brain tumors whose prognosis is strongly influenced by their extent of invasion into the surrounding brain tissue. While lower-grade gliomas present more circumscribed borders, high-grade gliomas are aggressive tumors with widespread brain infiltration and dissemination. Glioblastoma (GBM) is known for its high invasiveness and association with poor prognosis. Its low survival rate is due to the certainty of its recurrence, caused by microscopic brain infiltration which makes surgical eradication unattainable. New insights into GBM biology at the single-cell level have enabled the identification of mechanisms exploited by glioma cells for brain invasion. In this review, we explore the current understanding of several molecular pathways and mechanisms used by tumor cells to invade normal brain tissue. We address the intrinsic biological drivers of tumor cell invasion, by tackling how tumor cells interact with each other and with the tumor microenvironment (TME). We focus on the recently discovered neuronal niche in the TME, including local as well as distant neurons, contributing to glioma growth and invasion. We then address the mechanisms of invasion promoted by astrocytes and immune cells. Finally, we review the current literature on the therapeutic targeting of the molecular mechanisms of invasion.

## 1. Introduction

Gliomas are the most frequently encountered primary malignant central nervous system (CNS) tumors [1]. They are divided into four grades, characterized by different prognoses. Grade 1 and 2 gliomas are often localized and self-limiting, meaning that patients may strongly benefit from surgical resection. On the other hand, grade 3 and 4 gliomas are highly malignant and diffusely infiltrate the neighboring brain tissue [2]. Grade 4 gliomas, particularly isocitrate dehydrogenase (IDH)-mutant astrocytoma and glioblastoma (GBM), IDH-wildtype, bear the worst prognosis with a five-year survival of 4–6.7% and a ten-year survival of only 0.7% [3,4,5]. 

The main challenge in treating GBM is represented by tumor cell invasion into healthy brain tissue, which is the basis for subsequent tumor recurrence [6]. Studies have indicated that broader resection extents can help delay recurrence and increase overall survival rates [7,8,9,10]. However, gross-total surgical resection can be challenging to achieve due to the presence of brain structures that need to be preserved to maintain neurological function [11]. Additionally, even if a tumor appears to be entirely removed at the macroscopic level, microscopic eradication of infiltrating tumor cells is impossible and grants subsequent recurrence. This phenomenon is evidenced by post-mortem studies [12]. Therefore, understanding the mechanisms causing GBM cells to invade the brain tissue and resist treatment represent a cornerstone for future therapies in invasive, high-grade gliomas. This review aims to identify the molecular underpinnings of brain invasion in GBM, discussing molecular drivers in tumor cells and the microenvironment, as well as addressing the current therapeutic strategies against invasion mechanisms, from targeted molecular therapies to surgical and interventional solutions. 

## 2. Invasive and Proliferative Capacity of GBM Cells

### 2.1. Invasive and Proliferative Capacity of GBM Cells

In 2013, one study by The Cancer Genome Atlas (TCGA) group classified GBM into four subtypes based on transcriptional and mutational data: classical, neural, proneural, and mesenchymal [13]. A later study of single-cell analysis showed that these subtypes are mirrored in four transcriptional cell states: astrocyte (AC)-like, neural progenitor cell (NPC)-like, oligodendrocyte progenitor cell (OPC)-like, and mesenchymal (MES)-like, respectively. Each GBM tumor contains different proportions of each cell state, thereby conferring a leading subtype that dictates its clinical behavior [14]. While the first three states resemble normal brain cells at various stages of neurodevelopment, MES-like cells are more associated with injury response and immune cell infiltration. These four cell states interchange dynamically with one another and are influenced by both their genetic outlook and cues from the microenvironment [14,15,16]. The resulting cell state plasticity is responsible for the intra-tumoral and inter-tumoral heterogeneity that makes these tumors particularly elusive to treatment. Recent findings in the literature support the idea that the spatial organization of cell states in the tumor serves functional purposes of proliferation and invasion. Some studies have found differences in gene expression and mutation profiles between the non-enhanced central area and the enhanced borders of tumors when using magnetic resonance imaging (MRI)-guided samples [17,18]. Furthermore, cells at the invasive margins demonstrated a higher degree of transcriptional plasticity [19,20]. Venkataramani and colleagues [21] addressed these findings by combining microscopy and single-cell multi-omic techniques. They showed that the bulk of the tumor mass is composed mainly of MES-like and AC-like GBM cells [21], organized in a dense network connecting tumor cells with each other and with astrocytes through microtubes (Figure 1A). This network facilitates the propagation of transient intracellular Ca^2+^ currents (ICCs) that boost cell proliferation. Conversely, cells with neuronal-like cell states (NPC-like and OPC-like) are typically found at the tumor rim, unconnected to either the tumor network or astrocytes, migrating and invading the surrounding brain tissue. Their gene expression and phenotypic features, influenced by their neuronal states, mirror those of migratory neuronal progenitor cells during neurodevelopment [21]. These observations provide a mechanistic link between migration during brain development and the invasive capacities of these tumor cells. Importantly, invading cells receive synaptic inputs from neighboring neurons to increase their migration speed [21]. After microscopic dissemination and distant brain colonization, invasive GBM cells with neuronal states start to interconnect with each other and with astrocytes, switching to AC-like/MES-like states to establish new cellular networks typically found in the primary tumor core [21]. Greenwald and colleagues [22] shared similar findings, showing that cells with the same cell state cluster together in microscopic areas, where the TME is uniquely adapted to sustain their specific cancer cell features. Hypoxic gradients appear to organize the structural continuum of cell states in the tumor tissue. In outer tumor layers and the invasive zone, the three neuro-developmental cell states (AC-like, NPC-like and OPC-like) interact preferentially with each other and with their normal counterparts: AC-like cells with astrocytes, NPC-like cells with neurons, OPC-like cells with oligodendrocytes [22]. Instead, in the tumor core, MES-like cells interact with each other and with immune cells. Importantly, a peculiar group of MES-like GBM cells features a gene expression associated with gap junctions and glioma microtubes, making them most suitable to form the tumor network [22]. This network conformation in the tumor core has several biological advantages for GBM. As Hausmann and colleagues [23] pointed out, the GBM network is scale-free and follows small-logic dynamics that speed up intercellular communication. The network better withstands chemical damage induced by temozolomide (TMZ) and regenerates its connections among remaining cells after surgery by increasing new microtube formation [24,25]. The hubs of this network are called “periodic cells”, representing around 4% of all GBM tumor cells [23]. They possess a MES-like cell state and initiate ICCs. These currents, generated by the overexpression of the KCa3.1 channel (a potassium-regulated calcium channel), boost tumor proliferation by activating the MAPK and NF-kB pathways [23], and regulate microtube dynamics for networking and invasion. The selective ablation of hub cells impaired the overall tumor growth, although this impairment was counterbalanced by a cancer cell change in identity from non-periodic to periodic [23]. This shows yet another level of phenotypic plasticity that may contribute to therapeutic resistance. Interestingly, Minata and colleagues [20] have identified a proneural-to-mesenchymal shift in invasive GSC signatures upon exposure to ionizing radiation (IR) [20]. MES-like glioma stem cells (GSCs), typically localized at the tumor core, may be selectively induced by IR through activation of the c/EBP-β signaling, a finding that could help resolve previous reports of mesenchymal tumor phenotypes at the invasive edges of gliomas [20,21]. 

Overall, these findings shed light on the cellular and structural architecture of GBM tissue. The distribution and localization of tumor cell states has important functional significance: GBM cells with neuronal-like states are those primarily involved in glioma invasion and dissemination, while MES-like GBM cells form networks adapted for enhanced communication and proliferation through ICC propagation. The transcription factor OLIG2 facilitates the migration of NPC-like and OPC-like tumor cells along vessels and white matter tracts [26,27,28]. In fact, OLIG2 triggers Wnt7b and CXCR4, which promote the maintenance of a progenitor-like state and cell adherence to the vasculature, respectively [29,30]. When cell migration is terminated and tumor cells start to detach from the perivascular niche, OLIG2 is downregulated, and cells shift towards a MES-like state [31]. 

The concept of the core tumor mass behaving as excitable and authentic syncytial tissue has profound implications for the field of brain cancer biology, given their recrudescence and endurance against conventional treatment strategies. Furthermore, identifying microscopic areas with invasive features or immune cell infiltration will be pivotal when considering the development and the efficacy of future therapeutic strategies in GBM.

### 2.2. Tunneling Nanotubes and Tumor Microtubes in Glioma

Glioma cells display two types of cellular projections: tumor microtubes (TMs) and tunneling nanotubes (TNTs) (Figure 1B). TMs and TNTs connect GBM cells to each other to form cell networks and to the astrocytic and neuronal niche in the TME [32]. Tunneling nanotubes (TNTs) are thinner and transient, and their contacts are either open-ended or mediated by gap junctions [33,34]. Composed of mainly actin and microtubules [35], these connections allow for the exchange of small molecules, calcium currents, endosomes, and organelles such as the endoplasmic reticulum, Golgi apparatus, lysosomes, and mitochondria [36,37]. In addition to facilitating tumor-tumor and neuron-tumor contacts, the exchange of molecules through TNTs can promote invasiveness, metabolic plasticity, proliferation, resistance to treatment, and angiogenesis [37]. 

Alternatively, tumor microtubes (TMs) are longer and thicker than TNTs and perform connections through gap junctions and adherens junctions [24,38]. They bear physical similarities to neurite growth cones [24,39] and share some of their structural proteins including growth-associated protein 43 (GAP43) [40], Tweety Homolog 1 (TTYH1) [41,42,43], catenin-delta 1 (CTNND1) [38], and transforming growth factor-beta (TGF-β) [44]. In the context of GBM, tumor cells exploit TMs to interact with each other and with the microenvironment [24,38,41,45]. Within the GBM cell network, connecting TMs enable the transfer of calcium, ATP, AMP, inositol triphosphate, and small molecules under 760 kDa [37]. Calcium ions are especially important in the context of tumor growth and therapeutic resistance, as their redistribution across the tumor network enables gliomas to tolerate the oxidative damage and the apoptotic calcium-mediated signals induced by radiotherapy [46]. Furthermore, the spreading of ICCs across the network serves as a triggering signal for tumor proliferation [23]. Microtubes also enable the transfer of mitochondria from astrocytes to GBM cells through GAP43, a major TM structural protein [47]. Mitochondrial transfer induces metabolic reprogramming within GBM cells, triggering their proliferation and enhancing their stem-like features [47]. Unconnected GBM cells, alternatively, display blind-ending, invasive TMs with higher turnover and movement patterns like protrusion, retraction, or branching [21]. These movements are reminiscent of those occurring in migratory neurons [48,49], reflecting the neuronal origin of invasive GBM cells [21]. In this case, microtubes mediate the synaptic contacts of glioma cells with neurons, which resemble those occurring between migrating immature neurons and OPCs during neurodevelopment [50,51]. Bipolar GBM cells with one or two TTYH1+ TMs are more migratory and invasive than multipolar cells featuring more than four TMs [21,29,41,52]. This is because in bipolar tumor cells, TTYH1 colocalizes with chloride-ion channels and integrin-α5 molecules, which control cell volume and navigation through neuropil and the ECM [27,39,52,53]. Moreover, TTYH1 knockdown caused formation of dysfunctional TMs that exhibited poor invasion and proliferation, suggesting that TTYH1 drives tumor invasiveness but not connectivity [36,37]. 

Neuronal activity via α-amino-3-hydroxy-5-methyl-4-isoxazole propionic acid receptor (AMPAR)-mediated calcium currents further increases microtube branching, turnover and average step length in unconnected GBM cells by activating CREB signaling [21]. These mechanisms effectively speed up glioma invasion of brain tissue. Therapeutic targeting of AMPAR-mediated neurotransmission through the anti-epileptic drug perampanel significantly reduced TM formation and branching in mice models.

Previous studies have indicated the sheer presence of microtubes as a marker of malignancy [46]. Higher gap junction expression in TM-connected glioma networks is associated with higher cell stemness compared to unconnected cells [54,55]. Moreover, astrocytomas feature longer and more abundant TMs compared to other lesions with a more benign behavior, such as oligodendrogliomas. Microtubes appeared to be longer in higher grades of astrocytoma than in lower grades, and tumor-tumor networks were rarer in oligodendroglioma than in astrocytoma [24]. This could be related to the fact that 1p-19q co-deletion in oligodendroglioma involves genes encoding the essential proteins in the formation of TMs, including TTYH1, neurotropic growth factor (NGF), and neurotrophin 4 (NT4), which promotes GAP43 expression [24,25,38,56]. These findings demonstrate how TMs and TNTs are important mediators of tumorigenicity and invasion in gliomas, and targeting their regulatory mechanisms may represent an important avenue in future therapeutic strategies. 

## 3. The Extracellular Matrix in GBM Invasion

One of the main clinical features of high-grade glioma is the significant spread of the tumor cells into healthy brain tissue. GBM can invade the area surrounding the tumor in two ways: collectively as a strand of cells or diffusely as single cells [57]. 

Collective invasion in GBM occurs in a specific direction and can lead to increased neurotoxicity and seizures, which is one of the first presenting symptoms. It can also cause the tumor to spread more quickly into the surrounding brain [30,38,57,58]. 

In this context, the ECM may work as an obstacle to cell invasion or as a scaffold supporting the invading cells’ growth [56]. Indeed, ECM components have a fundamental role in diffuse tumor cell infiltration [59]. Hyaluronan (HA), one of the primary brain ECM components, seems to facilitate cell migration by binding to its cognate receptor, CD44 [60,61,62,63]. Similarly, osteopontin (OPN) acts as a CD44 ligand, triggering its intracellular signaling and CREB gene expression that is responsible for glioma cell perivascular migration and invasion and contributes to maintaining a stem-like phenotype [64,65]. Furthermore, fibronectin, which is highly enriched in mesenchymal-type gliomas, stimulates collective invasion by increasing the cohesion of GBM cells [65,66], while collagen molecules in the perivascular niche promote tumor invasiveness through upregulation of integrin and PI3K/Akt signaling [59,67,68]. Tumors also upregulate several different integrins to stimulate migration and dynamic interactions of the cytoskeleton and the ECM [66,69,70,71]. Finally, there is evidence of the involvement of different laminin isoforms in promoting glioma invasion. Laminin-2 and laminin-5 are preferentially upregulated in invasive GBM areas, while laminin-8 silencing has resulted in significant impairment of GBM invasion [65,72,73,74]. 

Gliomas also have higher metalloproteases (MMP)2 and MMP9 expression compared to normal brain tissue [65,75,76]. Generally, MMPs are released by either the tumor cells or the brain parenchyma. These specific MMPs regulate ECM integrity and activate cytokines in the interleukin-1 (IL1) and interleukin-8 (IL8) families, which control growth and invasion [77]. In particular, MMP2 is involved in the migration through collagen type I, which is present in the core of the tumor and the perivascular space [6,59]. 

## 4. The Neuronal Niche in Glioma Invasion

### 4.1. The Neuron-Glioma Interface: Neurogliomal Synapses

Glioma cells establish synapses with adjacent or distant neurons through their tumor microtubes [45,78]. These neuroglioma synapses (NGS), with neurons on the presynaptic side and GBM cells on the postsynaptic side [45,79], boost tumor cell proliferation, malignant transformation, and early invasion [78]. Furthermore, NGS enable glioma tissue integration within neural circuits, altering their plasticity and clinical function [80]. At the NGS level, both direct neurotransmitter stimulation and paracrine release of synaptic factors stimulate glioma cells, with consequent proliferative and invasive effects (Figure 1C). Approximately 10–30% of adult and pediatric gliomas are characterized by neuron-tumor glutamatergic networks. These connections are particularly enriched in GBM and H3 K27M-mutated diffuse midline gliomas, as demonstrated by both human tissue sections and animal models [45,79]. 

#### 4.1.1. Direct Synaptic Stimulation in Glioma Compared to Brain Metastases

Venkataramani and colleagues [45] showed that GBM cells with neuronal cell states, located in the infiltrative zone, engage in unilateral glutamatergic synapses through their microtubes. These synapses are mediated by AMPARs and induce calcium-based excitatory post-synaptic potentials (ePSPs) in glioma cells, boosting microtube dynamics, microtube turnover, invasion speed, and proliferation [21,23,37,41,45,55,79]. Other mechanisms of activity-dependent glioma stimulation include slow inward Ca^2+^ currents (SICs), which are regulated by an admixture of AMPAR, inward potassium currents, glutamate transporters, and gap junctions in different proportions among tumor cells [45,79]. The functional role of SICs requires further elucidation, but this finding highlights the heterogeneous molecular contribution to the neuron-induced Ca^2+^ conductivity within GBM, a crucial drive of glioma progression. Indeed, the discovery of NGS uncovered a new and underappreciated mechanism of GBM growth. Therapeutic targeting of synaptic stimuli to glioma and metastatic cancer cells has promising implications in the neuro-oncology field. 

A different form of NGS has also been identified in brain metastases. Zeng et al. [81] described synaptic contacts in metastatic breast cancer cells (MBCCs), that overexpress the GluN2B NMDA receptor and engage in a synaptic liaison with pre- and post-synaptic neurons. MBCCs produce glutamate in an autocrine fashion, but not in sufficient amounts to stimulate their own GluN2B receptors. Therefore, metastatic tumor cells supplant astrocytes at the level of synaptic clefts to expose themselves to synaptic glutamate and ensure their own glutamatergic stimulation. The resulting NMDA-mediated tumor cell activation enhances both proliferation and colonization of the brain. Previous work by Chen and colleagues [82] showed that breast and non-small cell lung cancer (NSCLC) cells connect to astrocytes via gap junctions made of connexin 43, transferring cyclic GMP-AMP (cGAMP) to astrocytes to trigger the release of IFNα and TNF. In turn, these pro-inflammatory molecules induce STAT1 and NF-kB signaling in tumor cells to trigger proliferation and chemoresistance. Finally, Savchuk et al. [83] showed that small-cell lung cancer (SCLC), a highly malignant tumor often metastatic to the CNS, receives synaptic stimulation from cortical neurons after colonizing the brain, and its cells undergo calcium-dependent depolarization sufficient to trigger proliferation. In turn, SCLC cells switch to an astrocyte-like phenotype and induce neuron hyperexcitability, pointing to complex reciprocal interactions at the tumor-neuron interface [83].

#### 4.1.2. Paracrine Factor Release from Neurons Promotes Glioma Growth

Apart from the direct stimulation mechanism of NGS, glioma cells also exploit a plethora of proteins, released by neurons in the TME in a paracrine fashion, for their growth and invasion. For paracrine NGS, a bidirectional interaction is established with local and distant neurons to foster the release of these signals. The net effect is a cycle of excitation and stimulation of the tumor tissue, favoring glioma growth, integration in the brain network, and hyperexcitability which fosters further proliferative signals. 

In 2015, Venkatesh et al. described how the paracrine release of neuroligin 3 (NLGN3) from neurons had mitogenic effects on gliomas [84]. NLGN3 is a postsynaptic cell adhesion protein binding to its presynaptic cognate neurexin 1 (NRXN1). In glioma, NLGN3 is cleaved from synaptic neurons and OPCs in an activity-dependent fashion by the ADAM10 metalloproteinase [85]. Once released, soluble NLGN3 (sNLGN3) diffuses to glioma cells and binds surface receptors, inducing the FAK-PI3K-mTOR signaling pathway that results in glioma cell proliferation [32,84,85,86]. At the gene expression level, NLGN3 induces the upregulation of TTYH1 gene, which enhances synapse-associated gene expression, tumor microtube formation and growth [39]. From a prognostic standpoint instead, high NLGN3 levels in HGG samples have been correlated with poorer patient survival [84]. Additional paracrine-released proteins with effects similar to NLGN3 include brain-derived neurotrophic factor (BDNF) and glucose-regulated protein 78 (GRP78) [84]. Notably, BDNF increased NGS number and promoted AMPAR trafficking to the postsynaptic membrane in order to boost the strength of NGS signals [87]. These findings were substantiated in a recent article by Taylor and colleagues [87], who analyzed GBM and diffuse midline glioma (DMG) samples. They showed that the increase in postsynaptic AMPAR trafficking at the NGS level promoted by paracrine BDNF action is regulated by the tyrosine kinase B- calmodulin kinase II (TrKB-CaMKII) signaling. Moreover, they compared the effects of NLGN3 and BDNF on glioma, reporting that while NLGN3 upregulates the expression of AMPAR subunits, BDNF-TrKB signaling promotes AMPAR subunit trafficking to the glioma cell membrane [87]. Interestingly, this pathway modulates both the number and strength of direct neuron-to-glioma synapses, and the resulting synaptic plasticity is reminiscent to the one of healthy neuronal circuits during learning and memory processes [87]. 

In a different instance of paracrine neuronal stimulation, callosal projecting neurons (CPNs) from the contralateral unaffected brain hemisphere appear to favor glioma growth, early malignant transformation, and invasion by releasing semaphorin-4F (SEMA4F) in the local TME [78]. Most CPNs originate from layer 2 and 3 of the contralateral neocortex and interact with a subset of GBM cells that are present at the tumor rim and enriched in axon guidance and synaptic genes. The paracrine release of SEMA4F by CPNs remodels local circuits to reduce inhibitory synapses and favor network hyperactivity. While local ipsilateral neurons can induce cell proliferation themselves, CPNs in the contralateral hemisphere are also able to induce precocious, activity-dependent tumor infiltration and migration across the brain [78]. Single-cell RNA sequencing revealed an increase in glutamatergic synapse and axon guidance genes in GBM cells upon SEMA4F stimulation. Additionally, CPN stimulation and SEMA4F secretion were responsible for malignant transformation in low-grade glioma (LGG) models. 

#### 4.1.3. Paracrine Factor Release by Tumor Cells Boosts Hyperactivity and Malignant Neuroplasticity

Tumor cells themselves may release paracrine factors to favor their growth by stimulating surrounding neurons, resulting in network hyperactivation and seizures. Yu and colleagues describe the role of glypican 3 (GPC3) as a driver for both gliomagenesis and hyperexcitability [88]. GBM tumors with high GPC3 expression exhibited driver mutation variants in *PIK3CA* gene and induced intense brain remodeling during their growth, inducing hyperexcitation and epileptogenesis [88]. 

Another paracrine promoter of glioma progression is a glycoprotein called thrombospondin 1 (TSP1), released mostly by GBM cells but also from astrocytes and microglia [80]. In physiologic conditions, TSP1 regulates NPC differentiation and proliferation [89]. In the malignant context instead, TSP1 fosters tumor cell integration and reorganization of local neural networks towards hyperexcitation. In this way, TSP1 endows GBMs with high functional connectivity (HFC) at the molecular and network level. Under the influence of paracrine TSP1 secretion, HFC glioma cells show increased microtube length and invasiveness. TSP1 was also detected in the serum of patients with GBM, and its levels directly correlated with the overall connectivity of the tumor itself [80]. 

Electrocorticography (ECoG) analysis on patients with GBM showed cortical hyperexcitation in tumor-infiltrated cortices [80]. Biopsies of tumor areas displaying HFC at ECoG revealed increased structural connectivity, in the form of increased postsynaptic puncta density, puncta cluster size and puncta colocalization at histological analysis. Neurons in these regions showed overactivity and increased network synchrony. Tumors with HFC voxels had a significant decrease in overall survival after stratification for other independent prognostic factors [80]. 

It appears that the neuronal integration of glioma tissue, as demonstrated by molecular and instrumental biomarkers, is a crucial indicator of tumor malignancy and poor prognosis. Drexler et al. have characterized the biology of tumors with neural signature and HFC [90]. These tumors displayed a peculiar epigenetic pattern with hypomethylated CpG islands, and gene expression enriched for synaptic genes and neuronal differentiation. They also featured an abundance of NPC-like and OPC-like cells with stem-like features and a significantly lower immune cell infiltration, mirroring recent findings showing marked local immunosuppression in areas of GBM tissue predominantly composed of neuronal-like GBM cells [91]. These “neural-high” GBMs exhibit robustly increased tumor connectivity when imaged using magnetoencephalography and reduced Gd-enhancing volumes in post-contrast T1 images. Importantly, gross-total resection (GTR) or near-GTR did not produce a significant survival benefit in neural-high GBMs, in contrast to “neural-low” tumors. 

Taken together, these studies show the importance of neuronal inputs in GBM progression. Both unilateral-direct and bilateral- paracrine synapses are pivotal to promote glioma cell proliferation and migration. The concept of neuronal niche is also extended beyond local neurons and encompasses also distant cortical cells in the unaffected hemisphere. This also shows the dramatic effect of GBM at the whole CNS level: neuronal stimulation allows the tumor to integrate into brain networks and change their function altogether, with consequent effects on patient survival and cognitive function [78]. 

The prognostic validation of synaptic and paracrine factor levels in GBM samples requires further characterization. Nonetheless, they may be important biomarkers in early tumor diagnosis given their presence in patient serum. They may also bear important information concerning tumor evolution and the resistance profile during the disease’s natural history.

Investigations into the therapeutic targeting of several of these mechanisms are underway, and results from several clinical trials should be forthcoming [92].

## 5. The Vascular Niche in Glioma Invasion

Blood vessels in the tumor tissue provide an important scaffold for tumor cell migration.

In this context, Farin et al. [71] co-cultured patient-derived GBM cells with organotypic brain slices. They showed that glioma cells adopted a unipolar stretched morphology with protrusions similar to those of glial progenitor cells along the vasculature. These protrusions, known as invadopodia or focal adhesions, are crucial for cell migration, highlighting the role of vasculature in glioma cell invasion [93,94]. Specifically, invadopodia can sense the extracellular matrix (ECM), break it down, and trigger cytoskeletal remodeling to propel the cell body forward and invade the surrounding parenchyma [95]. Intracellular calcium appears to support these cytoskeletal alterations, and an increased calcium concentration inside the cells may work with the transient receptor potential vanilloid type 4 (TRPV4) to activate migratory programs [96]. Yang and colleagues [97] have demonstrated that TRPV4 is localized at the invadopodia, and it can promote growth and invasion of GBM that has been implanted intracranially or subcutaneously in mice.

Moreover, endothelial cells (ECs) deploy several molecular mechanisms to favor GBM invasion. First, they secrete vascular endothelial growth factor (VEGF) which promotes glioma invasiveness along with neo-angiogenesis and trans-differentiation of GSCs into ECs [98,99]. Additionally, the soluble factor angiopoietin-1 (Ang-1) present in the perivascular niche binds Tie2 receptor on GBM cells, upregulating N-cadherin and integrin β1 to promote invasion [100]. The CXCR4-CXCL12 chemokine axis appears to regulate glioma invasion along the perivascular niche. In fact, its knockdown resulted in reduced growth, increased response to radiotherapy, and prolonged survival in preclinical murine and human models [65]. Murine models and patient-derived cultures have shown that GBM cells bind to EphrinB2 [101] on endothelial cells and can express bradykinin receptor 2 (BKR2) to bind to bradykinin. Bradykinin expression in endothelial cells usually increases during tumor progression [102,103].

The vascular niche also regulates OLIG2 gradients to promote migration of invasive neuronal-like GBM cells, as previously described. On the other hand, Olig2–tumor cells migrate via blood arteries as cell clusters [30]. Chemokine signaling is the primary driving force behind perivascular migration into brain parenchyma. In cell line orthotopic models, endothelial cells secrete IL8, which increases invasion and growth of glioma spheroids in a three-dimensional collagen matrix [104]. IL8 is vital for tumor growth as it can regulate the expression of stem cell markers in GBM and other malignancies [105]. Furthermore, endothelial cells upregulate NOTCH1 expression and signaling on glioma cells in the perivascular niche (PVN), thereby boosting their proliferation [41]. Conversely, NOTCH1 downregulation depleted tumor cells in the PVN but upregulated TM formation, networking, and plasticity, conferring resistance against radiotherapy-induced cytotoxicity [41]. This led to the hypothesis that NOTCH1 expression is responsible for the plastic switch between the PVN and the TM connectivity niche and the proliferative versus therapy-resistant features of GBM [41].

Glioma stem cells can become active thanks to IL8 signaling, leading to their invasive behavior [106]. IL8 activates NF-kB, which in turn mediates the activation of invasive pathways [107]. Moreover, the NF-kB pathway is activated when bradykinin binds to BKR2, promoting the migration of GBM cells and increasing intracellular calcium levels that assist in cytoskeletal remodeling [65,103,108]. These discoveries indicate that specialized GBM cells can travel through arteries in response to chemo-attractants released by endothelial cells.

Furthermore, the AMPA subunit of glutamate receptor 1 (GluR1) associates with 1-integrin at focal adhesions and is involved in adhesion to collagen, a major component of endothelial basement membranes and brain meninges [109]. This finding suggests that glutamate could promote the invasive behavior of GBM cells, as these cells have elevated intracellular calcium levels that could trigger glutamate exocytosis [56].

Finally, GBM cells adapt their migration even in response to anti-angiogenic treatment, by shifting towards perivascular glioma cell invasion, with phosphorylation and upregulation of Met receptor tyrosine kinase induced by VEGF ablation [110,111,112,113,114].

## 6. Astrocytes Promote and Support Invasion and Mesenchymal Transformation

Astrocytes display several mechanisms favoring glioma growth and invasion. First, astrocytes undergo reactive astrogliosis upon contact with invading GBM cells, releasing connective tissue growth factor (CTGF). This in turn binds the tyrosine kinase receptor A (TrKA) and integrin-β1 on GSCs, triggering NF-kB and zinc finger E-box binding homeobox 1 (ZEB1) activation which promote glioma cell infiltration [65,115]. Moreover, reactive astrocytes produce sonic hedgehog (SHH), which binds smoothened (Smo) and patched-1 (PTCH1) membrane protein on the glioma cell surface, promoting tumor growth, stemness and self-renewal [116,117]. Among the astrocyte-secreted molecules, IL6 enhances GBM cell invasion by upregulating MMP activity and favoring ECM, and glial cell-line derived neurotrophic factor (GDNF) promotes GBM cell invasion by upregulation of RET, MAPK, and PI3K/Akt signaling pathways [65,118].

Moreover, astrocytes support invasion and tumor growth in GBM through paracrine signaling [118,119,120]. Relying on the tumor-secreted receptor activator of NF-kB ligand (RANKL), the NF-kB pathway becomes upregulated in astrocytes. Consequently, astrocytes produce more transforming growth factor beta (TGF-β), which has been shown to promote invasion in murine models implanted with glioma cell line U87 xenografts [121,122]. Reactive astrocytes also produce pro-inflammatory cytokines like IL33, which fosters growth and dissemination of tumor cells by acting on the microenvironment. The microenvironmental remodeling is carried out through signaling involving the ST2 receptor [123]. Astrocytes also secrete MMP2 and MMP9 in response to IL33 signaling, which promotes GBM invasion and migration [119].

Another essential component that favors invasion is tenascin C. Gliomas are characterized by higher expression of tenascin C, which binds toll-like receptor 4 (TLR4) and induces M2 polarization of macrophages and microglia. This polarization is crucial in promoting dissemination, progression, and invasion [124,125,126,127]. Through the NF-kB pathway, tenascin C can also induce mesenchymal transition in tumor cells [128]. The NF-kB pathway is vital in promoting glioma invasion [56].

Hallal et al. [129] showed that GBM cells can condition the behavior of astrocytes by releasing extracellular vesicles (EVs) in the TME. Astrocytes then internalize EVs and undergo pro-tumorigenic changes in cytokine production to favor proliferation and invasion [129].

Other studies have reported that senescent astrocytes overproduce molecules such as hyaluronan, fibronectin, MMP2, and MMP9, which are all factors involved in invasion, NF-kB activation, cell survival, and migration [130,131,132]. They also secrete other cytokines implicated in the NF-kB and STAT3 pathways, such as IL6 and IL8. Activating these signaling pathways promotes stemness of GSCs, invasiveness [56], as well as temozolomide (TMZ) resistance [133]. Finally, senescent astrocytes are less capable of removing neurochemicals from the synaptic cleft, leading to excitotoxicity due to increased glutamate levels [134,135] which are exploited by cancer cells to further promote invasion [56].

## 7. The Immune Microenvironment in Glioma Invasion

The immune microenvironment was shown to have a role in glioma invasion. A summary of how microglia, neutrophils, and myeloid cells interact with glioma cells to promote invasion can be seen in Figure 2.

### 7.1. Microglia

Microglia display several mechanisms promoting glioma invasion. Specifically, GBM cells release microglia-attracting factors like CSF-1, IL-34, and stem cell factor (SCF). These molecules recruit and activate microglia, inducing them to release epidermal growth factor (EGF), which was shown to promote GBM migration in GL261 cell lines [136,137]. Concomitantly, GBM cells upregulate MMP2 and MMP14 which facilitate tumor invasion [136]. MMP14 works through a specific crosstalk mechanism in which glioma-associated microglia upregulate TLR2, and glioma cells increase the expression of TLR2 ligand versican [138]. Versican induces MMP14 expression through microglia, which favors tumor invasiveness [139]. This axis can be inhibited through the TLR2 inhibitor O-Vanillin [140].

Other molecules, including stromal-derived factor 1 (SDF-1), granulocyte monocyte-colony stimulating factor (GM-CSF), and epidermal growth factor receptor (EGFR) are secreted by microglia to facilitate glioma invasion [137]. In particular, EGFR increases the activity of serine and cysteine proteases as well as matrix metalloproteases to favor the breakdown of the extracellular matrix surrounding the tumor, augmenting cell migration in the brain tissue [141]. Finally, tumor invasion and aggressiveness decreased upon knockdown of prolyl 4-hydroxylase (P4H), an enzyme involved in collagen synthesis and secretion. This occurred concomitantly with microglia polarization towards an M1 pro-inflammatory phenotype [142]. Taken together, these findings highlight the implication of microglia in the process of glioma cell invasion.

### 7.2. Neutrophils

Neutrophils, a crucial component of the GBM TME, appear to play a role in tumor progression and invasion. Activated neutrophils secrete elastase, a protease that can degrade the extracellular matrix, potentially leading to the breakdown of healthy brain tissue surrounding the tumor and facilitating invasion [143]. A study conducted by Iwatsuki et al. [144] delved into the impact of elastase secretion by neutrophils on glioma progression. The study enrolled 12 patients who received surgical resection of glial tumors with various grades, including four GBM cases. The results show that elastase was notably absent from the tumor core in the GBM patients’ samples but enriched in the tumor edges. Meanwhile, lower-grade glioma samples exhibited reduced elastase levels, suggesting a potential link between neutrophil recruitment in GBM and enhanced invasion. Further studies need to be carried out to confirm or disprove this hypothesis.

### 7.3. Myeloid Cells

Myeloid cells include granulocytes and monocytes. Amongst myeloid cells, CD44-positive cells appear to have a role in glioma invasion. CD44, a cell surface molecule, is crucial for cell-to-cell and cell-to-matrix adhesion [145]. It is also implicated in the TLR2 signaling pathway and serves as a major regulator of MMP9 expression, which is involved in the IL33 signaling pathway and in glioma invasion, as previously described [119,146]. The absence of CD44 has a notable impact on MMP9 levels within the tumor microenvironment. In this context, functioning as a ubiquitous protein binding component of the extracellular environment, CD44 is overexpressed in various tumors, including gliomas [147]. Indeed, Tcyganov et al. [148] demonstrated that selectively removing CD44 from the myeloid cell population within the glioma tumor microenvironment led to a significant reduction in tumor invasiveness.

## 8. Therapeutic Targets of Glioma Invasion

### 8.1. Molecular Targeting

As novel mechanisms of glioma growth, invasion, and treatment resistance unfold, the next generation of targeted therapies should focus on obliterating these mechanisms and hindering the malignant cells’ ability to create tumor cell-tumor cell and tumor cell-neuron interactions. In this context, one of the tractable targets that have received wide interest are gap junctions. In fact, gap junctions elicit a significant role in the formation of tumor microtubes and maintaining the communication among tumor cells. In detail, blocking the function of gap junctions was found to disrupt the synchrony between glioblastoma cells that is mediated by intercellular calcium waves among them [21]. Consequently, several gap junction inhibitors have been investigated in the context of glioblastoma treatment and sensitizing tumor cells to chemoradiotherapy. For instance, the non-steroidal anti-inflammatory drug (NSAID) meclofenamate, which also has a gap junction inhibitory effect via blocking connexin-43, exhibited the ability to disrupt the tumor cell network and attenuate the tumor microtubule-mediated communication among malignant cells. Moreover, meclofenamate sensitized GBM cells to TMZ treatment and resulted in a significant reduction in tumor volumes of mouse models when added to oral TMZ treatment [55]. These encouraging findings inspired the initiation of the phase I/II clinical trial (MecMeth/NOA-24, registration number: EudraCT 2021-000708-39) investigating the safety/efficacy of meclofenamate and TMZ combination therapy in patients with first-time relapse of MGMT-methylated GBM [149]. Another interesting agent that has been explored in this context is INI-0602, which is a gap junction inhibitor that has been optimized to penetrate the blood brain barrier (BBB) and has shown promising effects in mouse models of Alzheimer’s disease and amyotrophic lateral sclerosis [150]. Although it did not show a significant effect on the viability of GBM cells on its own, INI-0602 was able to significantly augment TMZ’s anti-tumor effects in vitro [151]. Further in vivo studies are needed to confirm this sensitizing effect of INI-0602, as it offers an intriguing agent that can achieve the needed accumulation in the central nervous system. In addition to targeting gap junctions, disconnecting GBM cells can be achieved through targeting the dynamic assembly/disassembly of microtubules. In fact, several common chemotherapeutic agents elicit their function through targeting microtubules, and these include taxanes, vinca alkaloids, and colchicine. However, these agents face the obstacle of not being able to cross the BBB besides their cytotoxic systemic effects. Hence, a newly generated microtubule-targeting agent (ST-401) with CNS-penetrant abilities was developed and tested for its efficacy against glioblastoma. Mouse models treated with ST-401 achieved proper accumulation of the drug in the brain [152]. Moreover, the drug exhibited radio-sensitizing and chemo-sensitizing effects in a glioma mouse model and resulted in a significant survival benefit when added to either TMZ or radiation therapy [152].

An alternative strategy would entail disrupting the communication between neurons and glioma cells, which has been shown to promote proliferation, aggressiveness, and invasion. Hence, inhibiting the neuron-tumor cell crosstalk, both in its direct synaptic transmission-mediated and indirect paracrine signaling-mediated forms, can serve as potential mechanism to target glioma invasion. First, hindering the glutamatergic excitatory input that neurons transmit to glioma cells can abrogate the resulting activation of proliferation and invasion pathways. Based on this, antiepileptics attracted substantial interest in brain cancer therapy research. In particular, among several popular antiepileptics (levetiracetam, valproate, perampanel, and carbamazepine) that were investigated, perampanel, which is an AMPA receptor antagonist, exhibited superior anti-tumor effects on the six tested glioma cell lines [153]. This is in line with the findings that AMPA receptor-induced calcium signaling participates in the proliferative effects precipitated by glutamatergic transmission on glioma cells [154]. Further in vitro studies confirmed the anti-proliferative effect of perampanel on GBM cell lines as a single agent or as an add-on therapy on top of TMZ [45,155,156]. Interestingly, talampanel, another antiepileptic drug belonging to the same family of AMPA receptor inhibitors, has already been investigated in a phase II clinical trial and shown a survival benefit as an adjuvant to chemoradiotherapy compared to historic controls that received TMZ/radiotherapy only [157]. However, the clinical development of talampanel has been aborted due to its unfavorable pharmacologic characteristics, mainly its short half-life which necessitated frequent daily administration [32]. This highlights the promising potential for perampanel which shares the same mechanism of action but possesses a better pharmacologic profile. On the other hand, blocking the effects of paracrine signaling from neurons on glioma cells can also contribute to blunting their proliferative and invasive capabilities. This can be achieved by targeting neuroligin-3 (NLGN-3) and brain-derived neurotrophic factor (BDNF). Specifically, inhibition of NLGN-3 can be performed by preventing its release, which is mediated by the metalloprotease ADAM10. As expected, ADAM10 inhibitors, INCB7839 and GI254023X, significantly attenuated the growth of orthotopic xenografts of high-grade glioma [85]. In fact, INCB7839 is currently being investigated in a phase-I clinical trial involving pediatric patients with recurrent or progressive high-grade gliomas (NCT04295759). In a similar fashion, targeting TrkB, the receptor for BDNF, using the pan-Trk inhibitor entrectinib resulted in decreased tumor growth and increased survival in a diffuse intrinsic pontine glioma model (DIPG) [32].

Notably, thrombospondin 1 (TSP1) has been linked to both types of networks: tumor-tumor and neuron-tumor. In this context, tumor growth factor β (TGF-β) has been proven to promote the formation of tumor microtubes between GBM cells, and this function is mediated by TSP1 and SMAD [44]. Moreover, TSP1’s activation of its receptor the voltage-gated calcium channel α2δ-1 subunit drives synaptogenesis, which might contribute to the formation of connections between neurons and GBM cells [89]. Hence, targeting TSP1 exhibits a promising avenue for attenuating GBM invasion and resistance to treatment. Indeed, the chemotherapeutic drug apatinib, which is used in other types of cancer, exhibited the ability to attenuate the growth, migration, and invasion of human-derived glioma cells in vitro [158]. This effect is mediated by apatinib’s downregulation of the expression of the thrombospondin 1 gene (*THBS1*) [158]. In a similar fashion, inhibiting TSP1 using gabapentin resulted in decreased proliferation of GBM tumors in a mouse model [80]. These findings made the basis for an ongoing clinical trial investigating the role of glutamate inhibitors such as gabapentin, sulfasalazine, and memantine in addition to standard chemoradiotherapy in GBM (GLUGLIO, NCT05664464).

### 8.2. Targeting Ion Channels as Regulators of Cell Volume and Migration in Glioma

Ion channels are the foundation of cellular homeostasis and are vital to nervous system physiology [159]. Generally, actively proliferating cells are more depolarized than differentiated tissue [160], and this state of depolarization promotes proliferation. In fact, bioelectrical dysregulation can be both the cause and consequence of various neurological pathologies [161]. As such, the role of ion channels in driving glioma progression and invasion has been highlighted in several studies, and they appear as a plausible target to slow the dissemination of these tumors [162].

Moreover, ion channels appear to be crucially affected by electrotherapy approaches, such as tumor treating fields [163]. Glioma cell migration seems to be dependent on the electric field in a process called “galvanotaxis.” This phenomenon involves a migration flow in the anode’s direction when tissues are exposed to direct current electric fields. Interestingly, non-primary brain cancer cells that have metastasized from a different origin are unaffected by direct current stimulations, pointing toward a potential glioma-specific vulnerability [162]. Among ion channels, sodium leak channel non-selective protein (NALCN) seems to be a key player in the processes of cancer metastasis and nonmalignant cell dissemination [164].

Aquaporins (AQPs) and voltage-gated ion channels are two targets studied to limit glioblastoma migration. Various studies have demonstrated that AQPs 1, 4, and 9, as well as voltage-gated potassium, sodium, calcium, chloride, and acid-sensing ion channels are overactivated in GBM and can contribute to control of cell volume, degradation of extracellular matrix, rearrangement of the cytoskeleton, protrusion of lamellipodia and filopodia, and shifting of focal assembly sites and cell-cell adhesions [165,166]. Volume changes, primarily regulated by ion channels activity, are thought to be crucial in modulating tumor cell invasion in tight areas and the dynamics of cellular migration [167].

Pharmacological inhibition and shRNA-mediated knockdown of sodium-potassium-chloride co-transporter (NKCC) 1 expression appeared to reduce cell migration and invasion in vitro and in vivo. Interestingly, the knockdown of NKCC1 in glioma cells resulted in the formation of significantly more extensive focal adhesions and a 40% reduction in cell traction forces (i.e., forces responsible for maintenance of cell shape, cell motility, and cell communication [168]) in GBM cells compared to control cells [169].

During glioma invasion, cells move along the perivascular space and white matter tracts by propelling one side of the cell first, called the migrating edge, and then the opposite side, called the trailing edge [170]. In their review, Takayasu and colleagues conducted a detailed description of ion channels involved in glioma cell migration and invasion, highlighting six main channels involved in this process [170]. At the migrating edge, influx of Na+, K+, and Cl^−^ ions through the NKCC1 and concomitant K+ efflux through the K-activated Ca channel 1 (KCa1.1 or BK) generate an ionic gradient enabling water influx and filopodial swelling. In its turn, this causes the protrusion of these filopodia, which effectively starts the migration process [170,171,172,173,174,175]. In this context, BK inhibition was shown to downregulate GBM cell infiltration of brain tissue in both in vitro and in vivo models [171,174].

Concomitantly, the activation of TRP7 channel and volume-regulated chloride channel (VRAC) induces the expression of genes that promote a migratory phenotype and activation of the MAPK/ERK and PI3K/Akt pathways [170,176,177,178]. Subsequently, this results in promoting the activation of ion channels at the trailing edge, including KCa3.1 and voltage-activated chloride channel 3 (CLC3) [170,179]. The resulting efflux of K+ and Cl− ions favors water efflux and cell shrinkage at the trailing pole of the cell, resulting in cell retraction [170,179]. From the perspective of therapeutic targeting of ion channels in glioma invasion, silencing CLC-3 with siRNA can block cell migration [180]. Herein, phosphorylation by CaMKII serves as a regulatory mechanism for CLC-3 activity. Hence, the inhibition of both CaMKII and CLC-3 have demonstrated a reduction in the invasive potential of glioma cells expressing CLC-3 [181].

Additionally, 4-(2-Butyl-6,7-dichlor-2-cyclopentyl-indan-1-on-5-yl) oxybutyric acid (DCPIB) is a blocker of swelling-induced Cl currents, and it was shown to block the JAK2/STAT3 and PI3K/Akt signaling pathways, thus reducing invasion and proliferation [178]. However, DCPIB does not cross the BBB, so new analogs or the incorporation of innovative drug delivery techniques should be investigated in order to overcome the problem of BBB crossing [170].

There are several pharmacological treatments targeting ion channels, especially in cardiovascular pathology, and it was shown that patients receiving ion channel modulatory agents, such as verapamil, digoxin, amiodarone, or diltiazem had a decreased risk of developing glioblastoma as compared to patients not taking these medications (OR = 0.641 (*p* < 0.0001)) [182]. There is evidence of an essential role for ion channels in glioblastoma biology, especially in regulating cell volume and migration. However, more studies are required to understand how ion channel-targeted treatments can be incorporated into the clinical management of glioblastoma.

### 8.3. Targeting GBM Invasion and Integration through Brain Modulation and Surgery

Given the mounting evidence in the literature regarding the profound alterations of brain circuitry induced by GBM, along with their impacts on both cognitive performance and overall patient survival [80], future efforts may also tackle neural circuit impairments in glioma using interventional neurosurgical techniques. Procedures like focused ultrasound (FUS), deep brain stimulation (DBS), and transcranial magnetic stimulation (TMS) have been successfully implemented in selective circuit modulation to treat neuropsychiatric disorders such as obsessive-compulsive disorder, cognitive symptoms of Parkinson’s disease, tic disorder, addiction, and tremor [183]. While such approaches have yet to find applications in the field of neuro-oncology, it is possible to speculate that future efforts using selective, targeted circuit modulation procedures may help counteract the cognitive and functional deficits induced by glioma growth, integration, and invasion, thereby improving patients’ quality of life.

Furthermore, advances in neuroimaging and the study of the brain connectome can help highlight the structural and functional differences in brain circuitry throughout glioma growth and evolution. The completion of the Human Connectome Project (HCP) has resulted in the development of the Quicktome™ software (v1.1.1) [184], capable of single-patient identification of complex functional networks and subsequent integration into stereotactic surgery systems to increase the accuracy of intra- and extra-axial tumor resection [185,186]. Specifically, the Quicktome™ software overcomes the limitations of fMRI and DTI by employing reparcellation based on structural connectivity compared to anatomical connectivity, typical of DTI [187]. The software, through the use of DTI and contrast-enhanced T1 MRI sequences, allows for identification of the default mode network (DMN), salience network, language and limbic pathways, central executive network (CEN), dorsal (DAN) and ventral (VAN) attention network, and visual and sensorimotor networks [188,189]. This new tool may help assess alterations in brain networks induced by glioma, thereby guiding surgical interventions and modulation therapies. One significant aspect to consider in glioma surgery is tumor margin delineation, to differentiate healthy tissue from tumor tissue, edema, and artifacts. In high-grade gliomas, peritumoral infiltration and edema go hand in hand, complicating their distinction. Intra-operative two-dimensional ultrasonography (iUS B-mode) using a contrast agent (CEUS) allows for enhanced intraoperative distinction between tumor and healthy tissue, highlighting vascularized regions with proliferative activity, as opposed to necrotic or edematous areas [190]. However, it does not allow for adequate identification of circuits and networks. In parallel, FLAIR MRI sequences identify areas of peritumoral signal hyperintensity indicative of edema or possible infiltration but cannot distinguish between them. In this sense, other advanced (MRI) techniques could help differentiate tumor invasive borders from peritumoral edema. Price and Gillard [191] highlighted how different MRI techniques including diffusion MRI, perfusion MRI, and MR spectroscopy can provide useful information about glioma cellularity, angiogenesis, metabolism, and proliferation, respectively. Moreover, Witwer et al. [192] discussed how diffusion tensor imaging (DTI) parameters can successfully differentiate white matter (WM) disruption, infiltration, or displacement by glioma cells, as represented by variations in fractional anisotropy (FA) and apparent diffusion coefficient (ADC). Specifically, WM disruption is encountered as the absence of FA in the region, while tumor infiltration is signaled by a greater than 25% reduction in FA and ADC. Nonetheless, the same parameters apply for what is defined as WM edema. Conversely, mere WM fiber displacement is suspected in cases where FA reduction is less than 25%. One possible solution to the poor differentiation of glioma infiltration from surrounding edema is represented in the study by Zakharova and colleagues [193], who analyzed 50 cases of high-grade glioma and showed that diffusion kurtosis MRI biomarkers pinpointed structural tissue alterations in the brain tissue surrounding the tumor masses, delineating the invasive tumor borders in otherwise normally-appearing white matter. Moreover, Ruggiero and colleagues [194] conducted a preclinical evaluation of fast-field cycling nuclear magnetic resonance (FFC-NMR) and found that T1-relaxation at very low magnetic field through this sequence can successfully discern between proliferating and invasive/migrating glioma tissue, highlighting the promise of low-magnetic field relaxometry for future implementation in glioma patients. Finally, Hu et al. leveraged multi-parametric MRI techniques such as diffusion tensor imaging (DTI) and dynamics susceptibility contrast MRI (DSC-MRI) and combined it with spatially-matched multi-omic analysis to characterize the biology of invasive non-enhancing tumor borders [195]. This study shows the informative power of advanced multimodal imaging to study glioma invasion in humans at the structural and molecular level to better inform clinical decision making.

The latest efforts in glioma surgery seek to balance supramarginal tumor resection, meaning tumor resection beyond the contrast-enhancing borders and into the FLAIR-enhancing area, with the increased risk of neurological impairment [196]. Studies in the literature have suggested that supramarginal GBM resection may offer enhanced survival compared to standard gross-total resection (GTR), but these findings require further validation [10]. For this purpose, integration of advanced neuroimaging, intraoperative techniques, and even connectomic analysis of brain tumors may help identify invasive margins and network impairment, maximize the resection of both the tumor and infiltrated areas, and predict the chances of postoperative sensorimotor and cognitive deficits.

## 9. Personalized and Combination Therapies

Since GBM presents high intra- and inter-tumoral variability [15,197,198], it could be advantageous to target the uniqueness of each patient’s tumor profile [197]. Indeed, there are several different targets that could be the foundation of personalized glioma treatment (Table 1). However, despite the thorough genomic characterization, at present, a significant portion of these variants require further functional, biological, and clinical characterization. This lack of comprehensive characterization significantly challenges the prioritization of targets [197,199,200]. In the field of personalized therapy for GBM, tumor vaccines have been explored. With this technique, tumor antigens are used to stimulate a tumor-specific immune response, allowing to bypass the low mutational burden, immunological isolation, and immunosuppressive microenvironment, which are typical of GBM [201]. Unfortunately, no current study focuses on addressing GBM invasiveness in the light of personalized therapy approaches.

Combination therapy and drug synergism show potential in effectively addressing heterogeneous tumors and their interactions with the microenvironment. Table 2 presents an overview of combination treatment in GBM involving TMZ, radiotherapy, tyrosine kinase receptor (RTK) inhibitors, and immunotherapy [204].

In this context, some drugs that inhibit cell migration have been coupled with TMZ. A study by Park et al. [205] explores the role of coupling TMZ with chloride channel blocker 5-nitro-2-(3-phenylpropylamino)-benzoate (NPPB) to block glioma cell migration. Furthermore, a study by Chang et al. [206] explores the role of ethyl-1-(4-(2,3,3-trichloroacrylamide)phenyl)-5-(trifluoromethyl)-1H-pyrazole-4-carboxylate (Pyr3). Pyr3 is able to block both proliferation and invasion. By coupling it with TMZ, it is also possible to further hamper GBM growth. In 2021, Doan et al. [207] explored the use of a combinatorial drug regimen including TMZ, GPR-17 agonist 2-({5-[3-(Morpholine-4-sulfonyl)phenyl]-4-[4-(trifluoromethoxy)phenyl]-4H-1,2,4-triazol-3-yl}sulfanyl)-N-[4-(propan-2-yl)phenyl]acetamide, and alkylaminophenol against GBM. They saw that the combination of alkylaminophenol and GPR-17 agonist reduced GBM migration, invasion, and proliferation. Moreover, the addition of TMZ further enhanced the antitumor effect of the drug. Additionally, Jovanovich et al. [208] demonstrated that thioredoxin reductase inhibitors display anti-invasion potential in glioma and sensitize the cancer cells to TMZ treatment in culture. Finally, the GLUGLIO trial [209] studies the combination of TMZ with several glutamate inhibitors including gabapentin, memantine, sulfasalazine, and chemoradiotherapy. This phase Ib/II trial is ongoing and will yield results in 2026.

In a similar fashion, radiation induces a hyperinvasive phenotype in GBM. In a preclinical study by Stransky et al. [210], irradiation-induced hyper-invasion of glioma cells appears to be quenched by combining radiotherapy with TRAM-34, a drug targeting KCa3.1, in preclinical immunocompetent glioma mouse models. This finding is also supported by preclinical evidence from two studies by D’Alessandro et al., published in 2016 [215] and 2019 [216].

Other studies explore the use of combinations involving kinase inhibitors. Using an in vitro 3D collagen invasion assay, the combination of imatinib and docetaxel has demonstrated a significant reduction of glioma migration [211]. Furthermore, in a study by Hjelmeland et al. [212], combining LBT613, a RAF blocker, and everolimus, an immunomodulator, provided significant reduction in both the invasion and proliferation of glioma cells. Moreover, Jin et al. [213] showed that combining the Notch blocker MRK003 and the Akt blocker MK-2206 allows successful blockade of GBM cell invasion, although no effects are seen on cell proliferation.

For what concerns immunotherapy, Pilanc et al. [214] showed that combining an anti-PD1 antibody and arginase 1/2 inhibitor OAT-1746 in U87 and LN18 glioma cell lines appeared to potentiate PD1 blockade and inhibit tumor growth in glioma murine models. Furthermore, arginase 1/2 can have a direct inhibiting action on glioma invasion. The discussed combination therapies present potential avenues that should invite further investigation to facilitate their translation into clinical practice.

## 10. Conclusions and Future Directions

In this review, we tackled the various mechanisms that contribute to glioma invasion, focusing first on the intrinsic biologic drivers of glioma cell invasion and then shifting our attention to the various components of the tumor microenvironment. Table 1 provides a summary of the molecular pathways associated with invasion and the corresponding therapeutic targets, along with an overview of ongoing clinical trials in this field.

The neuronal niche has emerged as a new, crucial regulator and promoter of glioma proliferation and invasion, warranting future dedicated studies. The immune microenvironment in GBM has gained mounting attention in light of immunotherapy studies deploying cellular strategies and immune checkpoint inhibitors. Nonetheless, future studies should better define the way immune cells in the microenvironment can be co-opted to favor glioma invasion.

Despite many molecular targets being identified, the literature provides scarce evidence of therapeutic targeting of glioma invasion mechanisms, and most of these studies are at the pre-clinical stage. The most recent efforts leverage anti-epileptic drugs to quench abnormal activity-dependent glioma cell stimulation by the neuronal niche, and current studies should help define the effective clinical benefits of these strategies. Overall, invasion mechanisms represent a new frontier in our understanding of glioma biology and in developing new therapeutics, and successfully targeting these processes should help improve the survival and functional status of patients with GBM.

## Figures and Tables

**Figure 1 ijms-25-02563-f001:**
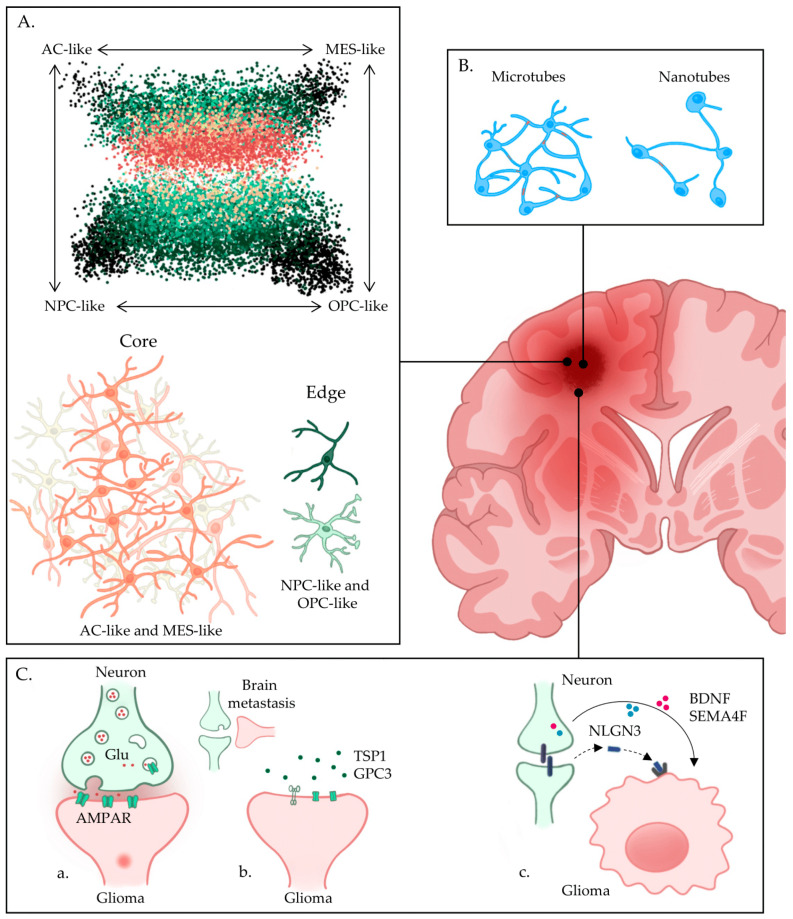
Intrinsic and Neuronal Drivers of Glioblastoma Invasion. (**A**) Intratumoral heterogeneity in GBM and functional localization of its cell states. MES-like and AC-like cells localize at the tumor core, while NPC-like and OPC-like cells are located at the invasive edge. (**B**) Tumor microtubes and tumor nanotubes allow tumor intercellular communication to enhance transport, excitability, and proliferation. (**C**) Neuron-glioma synapses enhance proliferation and invasion through three mechanisms: (**a**) direct neuron-glioma synapse and stimulation, (**b**) paracrine tumor cell release of neuronal stimulating factors, (**c**) paracrine neuronal release of tumor-stimulating factors. AC, astrocyte; AMPAR, α-amino-3-hydroxy-5-methyl-4-isoxazolepropionic acid receptor; BDNF, brain-derived neurotrophic factor; Glu, glutamate; GPC3, glypican 3; MES, mesenchymal; NLGN3, neuroligin 3; NPC, neural progenitor cell; OPC, oligodendrocyte progenitor cell; SEMA4F, semaphorin 4F; TSP1, thrombospondin 1.

**Figure 2 ijms-25-02563-f002:**
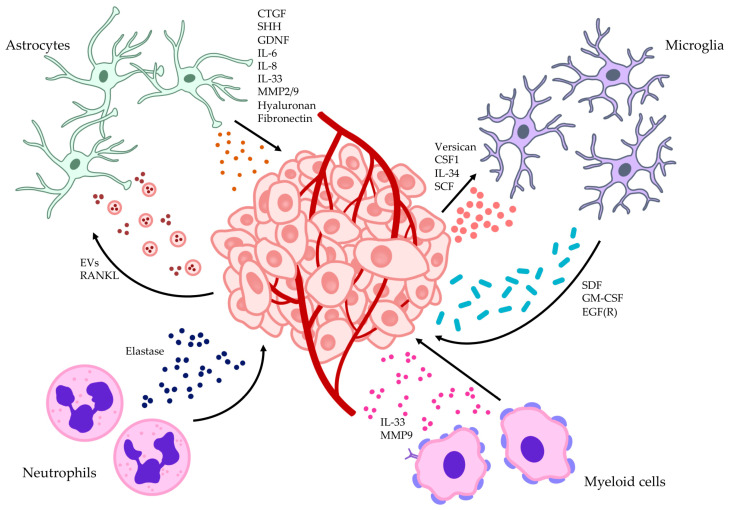
The astrocytic, vascular, and immune microenvironment contributions to glioblastoma invasion. Astrocytes and microglia reciprocally interact with the tumor by releasing factors promoting growth and invasion. Meanwhile, glioblastoma promotes astrocytic and microglial activation to favor its invasiveness and proliferation. Neutrophils promote extracellular matrix breakdown by secreting elastase, while myeloid cells upregulate CD44 and release cytokines and MMPs to facilitate glioma cell migration and dissemination. CSF, colony stimulating factor; CTGF, connective tissue growth factor; EGF, epidermal growth factor; EV, extracellular vesicles; GDNF, glial derived neurotrophic factor; IL, interleukin; MMP, matrix metalloprotease; RANKL, receptor activator of nuclear factor Kappa B ligand; SDF, stromal cell-derived factor; SHH, sonic hedgehog.

**Table 1 ijms-25-02563-t001:** Molecular targets and ongoing trials for invasion blockers in glioma.

Molecule/Molecular Axis	Targeting Agent	Clinical Trial
Targets on Glioma Cells		
KCa3.1 channel	Senicapoc	N/A
Microtubules	ST-401	N/A
OLIG2/Wntb7/CXCR4	N/A	N/A
Targets for Intracellular Currents in Glioma Cells		
GAP43	N/A	N/A
Connexin 43	Meclofenamate ± MZ	MecMeth/NOA24 [149]
	INI-0602 (connexin inhibitor)	N/A
TTYH1/integrin-α5	N/A	N/A
NGF	N/A	N/A
NT4	N/A	N/A
Targets in the Extracellular Matrix		
Osteopontin/CD44	N/A	N/A
Hyaluronan	N/A	N/A
Fibronectin	N/A	N/A
Collagen	N/A	N/A
Integrins	OS2966 (anti-β1 integrin monoclonal antibody)	NCT04608812 (terminated)
	[177Lu] Lu-FF58 (radioligand therapy against α-v-β-3/5 integrins)	NCT05977322 (recruiting)
Laminin 2/5/8	N/A	N/A
MMP2/MMP9	(GS) 5745, anti-MMP9 monoclonal antibody	NCT03631836 (unknown status)
Targets in the Neuronal Niche		
AMPAR	Talampanel	NCT00567592 [157]
BDNF BDNF-TrKB	Entrectinib	N/A
	pan-TrKB inhibitors	N/A
GRP78	N/A	N/A
NLGN3/ADAM10 (NLGN3 undruggable)	INC-B7839 and GI254023X (ADAM10 inhibitors)	NCT04295759 (recruiting)
SEMA4F	N/A	N/A
TSP1	Gabapentin Sulfasalazine Memantine	GLUGLIO (NCT05664464, recruiting)
	Apatinib	NCT04814329 (completed)
Targets in the Vascular Niche		
TRPV4	N/A	N/A
VEGF	Bevacizumab	[202]
VEGF	TMZ +/− Bevacizumab	GENOM 009 (NCT01102595, completed) [203]
Ang-1/Tie-2	Trebananib	NCT01609790 (completed)
CXCR4-CXCL12 axis	USL311	NCT02765165 (terminated)
NOTCH1	N/A	N/A
IL8	8R-70CAR	IMAPCT (NCT05353530, recruiting)
NF-kB/BKR2	N/A	N/A
GluR1	N/A	N/A
Targets on astrocytes		
CTGF	N/A	N/A
IL6	N/A	N/A
GDNF	N/A	N/A
RANKL	N/A	N/A
IL33	N/A	N/A
Tenascin C	Neuradiab™ (anti-tenascin C monoclonal antibody)	NCT00906516 (unknown status)
Targets on microglia		
EGF(R)	CM93 (small molecule anti-EGFR tyrosine kinase inhibitor)	NCT04933422 (not yet recruiting)
Versican/TLR2/MMP14	O-vanillin	N/A
SCDF-1	N/A	N/A
GM-CSF	N/A	N/A
EGF(R)	CM93 (small molecule anti-EGFR tyrosine kinase inhibitor)	NCT04933422 (not yet recruiting)
P4H	N/A	N/A
Targets on neutrophils		
Elastase	N/A	N/A
Targets on monocytes		
CD44	N/A	N/A
TLR2	N/A	N/A

KCa3.1: potassium-regulated calcium channel 3.1; GAP43: growth-associated protein 43; TMZ: temozolomide; TTYH1: Tweety Homolog 1; NGF: Nerve Growth Factor; NT4: neurotrophin 4; MMP2: matrix metalloprotease 2; MMP9: matrix metalloprotease 9; AMPAR: α-amino-3-hydroxy-5-methyl-4-isoxazole propionic acid receptor; BDNF: brain-derived neurotrophic factor; TrKB: tyrosine kinase B; GRP78: glucose-regulated protein 78; NLGN3: neuroligin 3; ADAM10: A Disintegrin and metalloproteinase domain-containing protein 10; SEMA4F: semaphoring 4F; TSP1: thrombospondin 1; GPC3: glypican 3; TRPV4: transient receptor potential vanilloid type 4; VEGF: vascular endothelial growth factor; Ang-1: angiopoietin 1; Tie-2: angiopoietin 1 receptor; CXCR4: CXC receptor 4; CXCL12: CXC ligand 12; IL8: interleukin 8; NF-kB: nuclear factor kappa B; BKR2: bradykinin receptor 2; GluR1: glutamate receptor 1; CTGF: connective tissue growth factor; IL6: interleukin 6; GDNF: glial-derived neurotrophic factor; RANKL: tumor-secreted receptor activator of NF-kB ligand; IL33: interleukin 33; EGF: epidermal growth factor; EGFR: epidermal growth factor receptor; TLR2: Toll-like receptor 2; MMP14: matrix metalloprotease 14; SDF-1: stromal-derived factor 1; GM-CSF: granulocyte-monocyte-colony stimulating factor; P4H: prolyl 4-hyroxylase; N/A: not applicable.

**Table 2 ijms-25-02563-t002:** Combination therapy strategies in glioma.

Molecule	Function	Combined Action	Reference
Temozolomide			
NPPB	Chloride channel blocker	Suppress cell movement	Park et al. 2016 [205]
Pyr3	Proliferation and invasion blocker	Inhibit GBM growth	Chang et al. 2018 [206]
GPR-17 blocker + Alkylaminophenol	Invasion and proliferation blockers	Inhibit proliferation	Doan et al. 2021 [207]
TrxR1 inhibitors 5/6	Thioredoxin reductase inhibitors and invasion blockade	Chemosensitization	Jovanovic et al. 2020 [208]
CRT ± Gabapentin Memantine Sulfasalazine	Apoptosis Invasion blockade through glutamate blockade	Increase survival	GLUGLIO trial Mastall et al. 2024 [209]
Radiotherapy			
TRAM-34	KCa3.1 inhibitor	Reduce radiotherapy-induced hyperinvasiveness	Stransky et al. 2023 [210]
Kinase inhibitors			
Imatinib + Docetaxel	c-Kit inhibitor Microtubule blocker	Reduce glioma invasion	Kinsella et al. 2011 [211]
LBT613 + Everolimus	RAF inhibitor mTOR inhibitor	Inhibit proliferation and invasion	Hjelmeland et al. 2007 [212]
MRK003 + MK-2206	Notch inhibitor Akt inhibitor	Suppress invasion	Jin et al. 2013 [213]
Immunotherapy			
Anti PD1 antibody + OAT-1746	Immune checkpoint inhibitor Arginase 1/2 inhibitor	Inhibit cell growth and invasion (U87 and LN18)	Pilanc et al. 2021 [214]

NPPB: 5-nitro-2-(3-phenylpropylamino)-benzoate; Pyr3: ethyl-1-(4-(2,3,3-trichloroacrylamide)phenyl)-5-(trifluoromethyl)-1H-pyrazole-4-carboxylate; GPR-17: 2-({5-[3-(Morpholine-4-sulfonyl)phenyl]-4-[4-(trifluoromethoxy)phenyl]-4H-1,2,4-triazol-3-yl}sulfanyl)-N-[4-(propan-2-yl)phenyl]acetamide; TrxR1: thioredoxin reductase; KCa3.1: potassium-regulated calcium channel 3.1; c-Kit: tyrosine-protein kinase KIT; RAF: rapidly accelerated fibrosarcoma; mTOR: mammalian target of rapamycin; Akt: protein kinase B.

## Data Availability

No new data were created or analyzed in this study. Data sharing is not applicable to this article.

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
