# Peer review of "Intrinsic and Microenvironmental Drivers of Glioblastoma Invasion"

_ijms, 2024, doi:10.3390/ijms25052563_

Round 1

Reviewer 1 Report

Comments and Suggestions for Authors

In this manuscript, Tyler et al. summarized and discussed the current understanding of molecular pathways and mechanisms used by tumor cells to invade normal brain tissue, which include insights into the biological drivers of invasion, interactions with the tumor microenvironment, and therapeutic targeting strategies. This is a comprehensive and detailed manuscript demonstrating the importance and potential of targeting tumor invasion as a therapeutic approach for glioblastoma. Therefore, I think this is an interesting and valuable manuscript. Publishing is recommended after the authors address the following issues.

1. The invading glioma cells undergo shape and volume changes. Some aspects of the energetic driving force, for example, ion channel-mediated energy drivers and drugs targeting these ion channels, are suggested to be discussed.

2. It is suggested to summarize the molecular signaling, pathways, and/or drugs associated with the invasion and neuronal interaction of gliomas.

3. There are some abbreviations in the text, such as TMZ in line 103 and GSC in line 113. It is recommended to include their full names for their first appearance.

4. Some references are missing Doi numbers.

Author Response

REVIEWER 1:

In this manuscript, Tyler et al. summarized and discussed the current understanding of molecular pathways and mechanisms used by tumor cells to invade normal brain tissue, which include insights into the biological drivers of invasion, interactions with the tumor microenvironment, and therapeutic targeting strategies. This is a comprehensive and detailed manuscript demonstrating the importance and potential of targeting tumor invasion as a therapeutic approach for glioblastoma. Therefore, I think this is an interesting and valuable manuscript. Publishing is recommended after the authors address the following issues.

Thank you for taking the time to read and review our manuscript and for your constructive feedback.

  1. The invading glioma cells undergo shape and volume changes. Some aspects of the energetic driving force, for example, ion channel-mediated energy drivers and drugs targeting these ion channels, are suggested to be discussed.

We thank the reviewer for the insightful suggestion. Shape and volume changes are indeed a driving force in glioma migration. We have added a specific sub-section (8.2. Targeting ion channels as regulators of cell volume and migration in glioma) on the topic, discussing the main ion channels regulating cell volume changes that drive glioma cell migration and potential ways to target them (lines 876- 943). 

  1. It is suggested to summarize the molecular signaling, pathways, and/or drugs associated with the invasion and neuronal interaction of gliomas.

We thank the reviewer for this suggestion. We have included a summary of the main molecular drivers and targets of glioma invasion and the currently available drugs, including those being investigated in clinical trials, in Table 1, which is now referenced in the text.

  1. There are some abbreviations in the text, such as TMZ in line 103 and GSC in line 113. It is recommended to include their full names for their first appearance.

Thank you for bringing this to our attention. We revised the manuscript to make sure that all abbreviations are defined at their first mention in the text.  

  1. Some references are missing Doi numbers.

We thank the reviewer for the comment. We have now addressed the reported issue.

Reviewer 2 Report

Comments and Suggestions for Authors

The manuscript entitled “Intrinsic and Microenvironmental Drivers of Glioblastoma Invasion” by De Fazio, E.; et al is a review work that appears to provide an in-depth overview of the biology of Glioblastoma, examining the molecular pathways and mechanisms involved in the invasion of tumor cells into normal brain tissue. In particular, the focus on the neuronal niche in the tumor microenvironment and the examination of interactions with astrocytes and immune cells add interesting perspectives. Reviewing the literature on therapeutic targets of molecular pathways of invasion is useful for understanding treatment possibilities.

To improve the manuscript the authors should consider including further insights into recent clinical trials regarding innovative therapies for Glioblastoma. Furthermore, it might be interesting to explore advanced diagnostic perspectives, such as the use of molecular biomarkers or imaging, for better identification and monitoring of tumor lesions. Integrating data on potential combination therapies or personalized therapies based on the patient's molecular profile could offer a more complete view of emerging therapeutic strategies. Finally, considering the inclusion of research that explores the implication of therapies targeting specific microenvironments within the brain could broaden the understanding of the complex interactions involved in Glioblastoma progression.

Furthermore, the information reported in paragraphs 3, 4, 5 and 6 could be organized in a table that could help readers to understand in a more concise and structured way the specific role of each component in the invasion of Glioblastoma.

For example, for paragraph 3, "The extracellular matrix in GBM invasion" a possible table could include information such as: ECM Component, Role in GBM Invasion, Specifics in Glioma Progression.

Author Response

REVIEWER 2:

The manuscript entitled “Intrinsic and Microenvironmental Drivers of Glioblastoma Invasion” by De Fazio, E.; et al is a review work that appears to provide an in-depth overview of the biology of Glioblastoma, examining the molecular pathways and mechanisms involved in the invasion of tumor cells into normal brain tissue. In particular, the focus on the neuronal niche in the tumor microenvironment and the examination of interactions with astrocytes and immune cells add interesting perspectives. Reviewing the literature on therapeutic targets of molecular pathways of invasion is useful for understanding treatment possibilities.

Thank you for taking the time to review our manuscript and for your valuable suggestions. We appreciate the constructive feedback provided.

To improve the manuscript the authors should consider including further insights into recent clinical trials regarding innovative therapies for Glioblastoma. Furthermore, it might be interesting to explore advanced diagnostic perspectives, such as the use of molecular biomarkers or imaging, for better identification and monitoring of tumor lesions. Integrating data on potential combination therapies or personalized therapies based on the patient's molecular profile could offer a more complete view of emerging therapeutic strategies. Finally, considering the inclusion of research that explores the implication of therapies targeting specific microenvironments within the brain could broaden the understanding of the complex interactions involved in Glioblastoma progression.

We thank the reviewer for the valuable and insightful suggestion. Indeed, a deeper analysis of the literature allowed us to identify a series of preclinical studies about the use of personalized and combination therapies for specific targeting of glioma migration and invasion. We have included these studies in a separate section (9. Personalized and combination therapies, lines 1219-1290) of the paper and summarized their content in Table 2, now referenced in the manuscript. We have also included a summary of the clinical trials that investigate the mentioned therapeutic agents in tables 1 and 2 to further highlight the translational potential on these agents. 

Furthermore, we have expanded our section tackling imaging techniques and modalities to study glioma invasion in section 8 (lines 1183-1203).

Furthermore, the information reported in paragraphs 3, 4, 5 and 6 could be organized in a table that could help readers to understand in a more concise and structured way the specific role of each component in the invasion of Glioblastoma.

For example, for paragraph 3, "The extracellular matrix in GBM invasion" a possible table could include information such as: ECM Component, Role in GBM Invasion, Specifics in Glioma Progression.

We thank the reviewer for this suggestion. We previously proposed Table 1 as a summary of the molecular targets of invasion identified in each specific subsection of the niche and microenvironment of glioma. We selected all the targets that have a demonstrated contribution specifically to the migration and invasion capacity of glioma cells in healthy brain tissue, and we believe the table provides an adequate representation of the molecular landscape of these mechanisms. In addition, we have also added another table (Table 2) that highlights the potential of combination therapies in avoiding treatment resistance and augmenting the effect of existing therapies.

Round 2

Reviewer 2 Report

Comments and Suggestions for Authors

The manuscript has been significantly improved. The changes made helped clarify the scientific soundness of the article. Additionally, reviewers' comments and criticisms were addressed, improving the overall quality of the work. Therefore, I believe that the manuscript has been sufficiently improved to justify its publication.